# The Utility of Virtual Patient Simulations for Clinical Reasoning Education

**DOI:** 10.3390/ijerph17155325

**Published:** 2020-07-24

**Authors:** Takashi Watari, Yasuharu Tokuda, Meiko Owada, Kazumichi Onigata

**Affiliations:** 1Postgraduate Clinical Training Center, Shimane University Hospital, 89–1, Enya-cho, Izumo shi Shimane 693-8501, Japan; konigata@gmail.com; 2Okinawa Muribushi Project for Teaching Hospitals, Okinawa 901-2132, Japan; yasuharu.tokuda@gmail.com; 3Nursing Department, Toho University Hospital Omori Medical Center, Tokyo 143-8541, Japan; meiko.owada@ns.toho-u.ac.jp

**Keywords:** clinical reasoning, virtual reality simulation, symptomatology, education support

## Abstract

Virtual Patient Simulations (VPSs) have been cited as a novel learning strategy, but there is little evidence that VPSs yield improvements in clinical reasoning skills and medical knowledge. This study aimed to clarify the effectiveness of VPSs for improving clinical reasoning skills among medical students, and to compare improvements in knowledge or clinical reasoning skills relevant to specific clinical scenarios. We enrolled 210 fourth-year medical students in March 2017 and March 2018 to participate in a real-time pre-post experimental design conducted in a large lecture hall by using a clicker. A VPS program (^®^Body Interact, Portugal) was implemented for one two-hour class session using the same methodology during both years. A pre–post 20-item multiple-choice questionnaire (10 knowledge and 10 clinical reasoning items) was used to evaluate learning outcomes. A total of 169 students completed the program. Participants showed significant increases in average total post-test scores, both on knowledge items (pre-test: median = 5, mean = 4.78, 95% CI (4.55–5.01); post-test: median = 5, mean = 5.12, 95% CI (4.90–5.43); *p*-value = 0.003) and clinical reasoning items (pre-test: median = 5, mean = 5.3 95%, CI (4.98–5.58); post-test: median = 8, mean = 7.81, 95% CI (7.57–8.05); *p*-value < 0.001). Thus, VPS programs could help medical students improve their clinical decision-making skills without lecturer supervision.

## 1. Introduction

Traditional lectures in Japanese medical schools have been primarily conducted as didactic lectures in large classrooms, with little interactivity. According to a previous study, for digital native students, such one-way, passive lectures are not effective and their learning outcomes may be relatively low [1,2,3,4,5]. Further, other reports have indicated that future medical education will make advancements through the implementation of digital tools such as video, audio, and simulators [1,4,6]. In fact, since the 1990s, research has especially focused on the application of virtual simulation technology to medical education [7,8,9,10,11,12]. Currently, virtual simulation including virtual patient is helpful in pre-clinical education, which can now utilize 3D images to teach subjects such as anatomy and pathology [13,14], training for pediatric surgery and laparoscopy [15,16], bioethics [17], and tracheal intubation techniques [18]. The technology has already been applied in the virtual simulation of hearing and vision loss to enhance medical students’ empathy for elderly patients [19]. A systematic review reported that the use of virtual patients can more effectively improve medical students’ skills and achieve at least the same degree of knowledge as traditional methods. The findings suggest that skills can be improved in a targeted way. The improved skills include procedural skills, a mixture of procedural and team skills, and some clinical reasoning skills. Moreover, virtual patient simulations (VPSs) in abnormal clinical scenarios may provide an accelerated breakthrough in improving clinical reasoning education [20]. However, there is inadequate evidence of the usefulness of VPSs for improving clinical reasoning skills for undergraduate medical students without lecturer supervision [21,22]. To the best of our knowledge, few studies have used VPSs to comparatively measure which areas of relevant case knowledge and clinical reasoning skills in symptomatology lead to better outcomes. Thus, this study had two objectives: (1) to clarify the effectiveness of VPSs for developing clinical reasoning skills among medical students without lecturer supervision, and (2) to elucidate whether VPSs improve clinical reasoning skills or knowledge of a particular case. To address these objectives, we used a pre–post experimental design to study a VPS.

## 2. Materials and Methods

### 2.1. Study Setting and Participants

This pre–post study was conducted at the Shimane University School of Medicine, a national medical school in Japan. It took place during one two-hour introductory class in clinical clerkship education in March 2017 and March 2018. Medical school programs in Japan are six years in length, and the students who participated in the study were at the end of their fourth year of courses, having recently passed the Objective Structured Clinical Examination. About 105 students are enrolled in each year of this medical school, so we arranged to perform the exact intervention in the same setting for a total of 210 people over two years. Before the intervention, we explained to each possible candidate: (1) the research intent, (2) the contents of the class, (3) the expected effects, (4) that data collection would be fully anonymous and collected with the distributed clicker, (5) that participation was free and voluntary, (6) that there was no conflict of interest, and (7) that we would not use these data for student evaluation.

A total of 191 participants attended the classes in this study; 19 of the original candidates did not participate due to absence. Of these, we analyzed a total of 169 participants (88.5%) after excluding 13 participants who did not provide consent and 9 participants who had trouble connecting their clickers and left the class early. The answers were not divided according to sex, as the data were collected anonymously using randomly distributed clickers. However, in 2017 and 2018, the proportion of fourth-year female students was 38% and 36%, respectively.

### 2.2. Study Design

This was a single center pre–post study of the effects of VPSs on clinical reasoning. A VPS software program (^®^Body Interact, Coimbra, Portugal) was used as an experimental intervention in both two-hour classes using an experimental design [23]. As the VPS in this study was based on a screen rather than a headset, it can be considered a partial VPS. In this software (Figure 1), the virtual patient shows the dynamic pathophysiological response to user decisions. Participants can act as real clinicians, ordering almost any tests or treatments as needed [23]. We first provided guidance on study participation, confirming that the responses of the participants who consented would be collected using a real-time audience response system (^®^TurningPoint clicker, Tokyo, Japan), and that the study would exclude the data of the participants who subsequently withdrew consent [24]. An audience response system clicker was randomly distributed to each participant to answer the questions in real time (Figure 2). The data were transmitted and saved to the computer at the front of the room, ensuring anonymity and making it impossible to identify individuals. To evaluate the VPS learning outcomes, we prepared a 20-item multiple-choice question (MCQ) quiz, which included 10 knowledge items and 10 clinical reasoning items for each of two scenarios (Appendix A List of 20 MCQs). We categorized the questions into knowledge items and clinical reasoning items from the past questions of the Japanese National Medical Practitioners Qualifying Examination in Japan, which were relevant to each scenario. The items were randomly arranged and selected from the Japanese National Medical Practitioners Qualifying Examination. The same MCQ quiz was administered to participants pre-study and post-study to evaluate the VPS outcomes. We explained that the participants could not discuss the answers with others. The difference between the scores indicated what the participants had learned through the intervention.

Two scenarios were used: (1) a 55-year-old male with altered mental status, and (2) a 65-year-old male with acute chest pain. Since the simulation software was partially in English, a faculty member (the same for each class) performed some minimal translations into Japanese as necessary. Another faculty member guided the participants in how to operate the VPS; this faculty member was instructed to intervene as little as possible during the class, especially regarding knowledge and clinical reasoning skills that were directly relevant to the answers for the pre-post tests. For each scenario, participants were allotted 20 min to make a diagnosis and treat the patient. During this period, participants were required to conduct a medical interview, physical examination, interpretation of results, and treatment as if they were practicing physicians. Following each scenario, the participants were automatically presented with a summary of their decision-making. Participants’ pre-post test responses were automatically sent to the computer as a CSV file. To ensure uniform study conditions and reduce confounding factors as much as possible, we used the same large lecture hall and ensured that all participants had the same level of experience in both 2017 and 2018. This was meant to minimize differences in VPS usage time and maintain fairness. The same interventions and methodology were used for all participants.

### 2.3. Statistical Analyses

We used an interquartile range (IQR) along with the 25th and 75th percentiles to indicate skewed data. The Shapiro–Wilk test was implemented to examine the total score of pre-post test distributions. McNemar’s test was used for paired nominal data pre–post study with Bonferroni correction, while the Wilcoxon signed-rank test was employed for nonparametric repeated measurements with skewed continuous variables. All analyses were performed using Stata statistical software, version 14.0 (*Stata 14 Base Reference Manual*; Stata Corp., College Station, TX, USA). All tests were two-sided with a *p*-value < 0.05 (Bonferroni correction in Table 1; *p*-value < 0.0025), which was considered statistically significant.

### 2.4. Ethics

The study was conducted in accordance with the Declaration of Helsinki. The Ethics Committee of the Shimane University Hospital did not conduct a review for the following reasons: participants’ personal information was not made available, the research data were automatically converted electronically to nondiscriminating data, participants volunteered and provided informed consent, the safety of VPS research is well-established, VPS research is widely used, and there was no risk of harm to the participants.

## 3. Results

Compared with pre-test baseline scores, participating students showed significant increase in average total post-test scores (*p*-value < 0.001). The total pre-test scores showed a median of 10 IQR (8–12), a mean of 10.1, and 95% CI (9.6–10.5); the post-test scores showed a median of 13 IQR (11–15), a mean of 13.0, and 95% CI (12.6–13.4). Figure 3 presents histograms of the pre-test and post-test scores for the 169 participants. The pre-test scores showed a normal distribution, while the post-test scores after the VPS intervention skewed to the right (Shapiro–Wilk test, *p*-value < 0.022). Furthermore, no significant differences were observed for these fourth-year medical students between the years 2017 (pre-test scores: median = 10 IQR (8–11), mean = 9.7, 95% CI (9.1–10.3); post-test scores: median = 13, IQR (11–15), mean = 12.7, 95% CI (12.1–13.2)) and 2018 (pre-test scores: median = 10 IQR (9–12), mean = 10.4, 95% CI (9.9–10.3); post-test scores: median = 13 IQR (12–16), mean = 13.3, 95% CI (12.7–13.9)).

Table 1 shows that the 10 items related to knowledge (K) and the 10 items related to clinical reasoning (CR), comprising past questions of the Japanese National Examination for Physicians, were randomly arranged. We compared the correct answer rate of the pre-post tests and performed McNemar’s test for each quiz and Bonferroni correction for all 20 items. As a result, 7 of the 10 CR items showed a statistically significant increase in the correct answer rate, whereas only 4 of the 10 K items showed a significant increase. The rate of change between CR items (median = +22.8% IQR 11.8% to 39.1%, mean = 25.3, 95% CI (12.1–38.5)) was significantly higher than that between K items (median = +4.2% IQR −1.8% to 11.8%, mean = 3.84, 95% CI (−5.1 to 12.8)), with *p*-value < 0.008. Notably, for the CR items, although the pre-test correct answer rate was low, the correct answer rate for some post-test items increased by 50 points or more (i.e., items 1 and 14). The items with a large increase in the correct answer rate included a direct question regarding diagnosis of an altered mental status and management of acute chest pain. On the other hand, some items requiring simple knowledge showed a decrease in the percentage of correct answers.

Figure 4 presents box plots of the scores for the 10 CR items and the 10 K items. CR scores increased significantly (pre-test: median = 5, mean = 5.3 95%, CI (4.98–5.58); post-test: median = 8, mean = 7.81, 95% CI (7.57–8.05); *p*-value < 0.001). K item scores overall also increased statistically, though the median was unchanged (pre-test: median = 5, mean = 4.78, 95% CI (4.55–5.01); post-test: median = 5, mean = 5.12, 95% CI (4.90–5.43); *p*-value = 0.003).

## 4. Discussion

As expected, overall scores were higher after the intervention. This can be explained by the fact that the histogram of overall scores was significantly skewed towards the higher scores. However, as a general rule with certain educational interventions, total scores will always rise immediately afterwards. This study assessed CR outcomes by using a VPS to train participants on diagnosing two clinical scenarios. The results show that VPSs improve CR ability even when used to instruct a large group in a lecture hall. On the other hand, little improvement was observed in K item scores when teacher intervention and instruction were minimized. There are several possible causes for this result. First, even if the participants had already learned the information necessary to answer all K items, these items may be difficult for fourth-year students because they were taken from the Japanese National Medical Practitioners Qualifying Examination, and the students have not yet performed their clinical clerkships. Second, the information is necessary to answer K items encompassed medical knowledge only, and may have been difficult to learn based solely on VPS unless teachers explained the scenario. For example, it is challenging to learn to interpret either electrocardiograms of patients with chest pain or contrast CTs based on anatomical knowledge simply by engaging with a VR program and without reading a textbook or listening to a lecture. Third, improving knowledge item scores in a short period of time may be challenging without essential knowledge of pathology, pharmacology, physiology, and anatomy. Furthermore, the knowledge acquired in this pre-clinical stage may still be perceived as complex when based only on engagement with the VPS. For example, the scores for some of the K items decreased in the opposite direction (items 9 and 18). One possible reason may be that analyzing the chest pain and altered mental status cases through the VPS program caused learner bias due to confusion over each scenario, and the answers were pulled in an unexpected direction. On the other hand, the scores increased for problems related to CR regarding altered mental status and acute chest pain. This is because these scenarios are more closely related to the diagnostic process of listing differential diagnoses for chest pains and determining necessary tests, as well as initially ruling out hypoglycemia for the differential diagnosis of altered mental status. For these reasons, we believe that engaging with a VPS is more useful for learning CR compared to acquiring medical knowledge, and for students to attain clinical experience by repeatedly utilizing the VPS.

For today’s digital native students, we believe it is necessary to implement new learning methods that include video, music, YouTube, and social media, rather than traditional methods [1]. One previous study compared three educational methods for teaching CR: live discussion, watching a video of the discussion, and learning from a textbook [25]. Notably, immediately after the lesson, the students who participated in the live discussion had a statistically significant outcome. However, an evaluation two weeks later showed no significant difference in knowledge retention from watching the discussion video or from participating in the live discussion, and both methods were found to be more effective than textbook learning.

In addition, another randomized controlled trial showed that the authenticity of CR (that is, how similar it is to actual practical experience) in traditional pre-clerkship instruction might not be high, as measured against pre-clerkship and clerkship outcome measures [26].

In this study, since we implemented a nonimmersive simulation without a virtual reality (VR) head set, we cannot consider it a true VR simulation by narrow definition. However, we gathered the participants in a large lecture hall with a shared screen to ensure uniformity of participant experience and to minimize confounding factors as much as possible. A fully immersive VR simulation, using a VR headset, may have a higher educational effect than a large shared screen [27]. It is not easy, however, for all participants to simultaneously use a fully immersive VR simulation, and it currently requires a tremendous amount of money and high-tech equipment [28]. Moreover, another comparative study found no overall significant differences in efficiency of educational outcomes and participant satisfaction between a shared screen and a fully immersive virtual reality simulation [29].

Furthermore, the greatest value of VPS is that the patient is not required to be involved in student training, and thus there is no risk to the patient [2,8,30,31,32,33]. There are also other merits, such as the ability to repeat the learning experience until the educational goal is achieved. Of course, it is not possible to complete all medical instruction through simulation education. However, we believe that if the strengths of traditional educational methods are combined appropriately with the VPS, further educational benefits can be expected.

### Limitations

Although we attempted to ensure a uniform study setting, there are some limitations to our study. First, this is not an experiment comparing VPS to other teaching methods. For that reason, we cannot say whether it is better than traditional teaching methods, such as lectures and case discussions. This discussion point is important, and further comparative experiments need to be conducted. Second, the intervention should ideally be reassessed afterwards to ensure that clinical reasoning skills have indeed improved and have been retained. Further, we cannot say whether performing multiple VPS scenarios will actually develop sufficient clinical reasoning skills for a clinician. However, our study could not be followed up because the data were taken anonymously using a clicker. Third, this study was conducted in a Japanese university hospital. Since clinical medical education systems vary from country to country, there is a question of external validity. It is uncertain whether these results are applicable to other countries or institutions. Fourth, variations in the degree of participation may be possible due to the type of VPS used. Further, although we tried to ensure consistency in the study setting, there may have been some unavoidable differences, such as the individual students’ active participation, concentration, and screen visibility due to seating positions. Fifth, about 10% of students were absent since no participation incentives were provided, and this may have created a selection bias. Finally, no similar studies have been conducted, and the validity of the test questions is uncertain, because they were selected from the Japanese National Medical Practitioners Qualifying Examination.

## 5. Conclusions

Our study suggests that VPS programs are more effective for increasing CR scores than K scores among medical students. VPS software programs could thus help medical students improve their clinical decision-making skills with minimal supervision from lecturers. In summary, the widespread use of VPS software programs in clinical education could help maximize the effectiveness of medical school curriculum.

## Figures and Tables

**Figure 1 ijerph-17-05325-f001:**
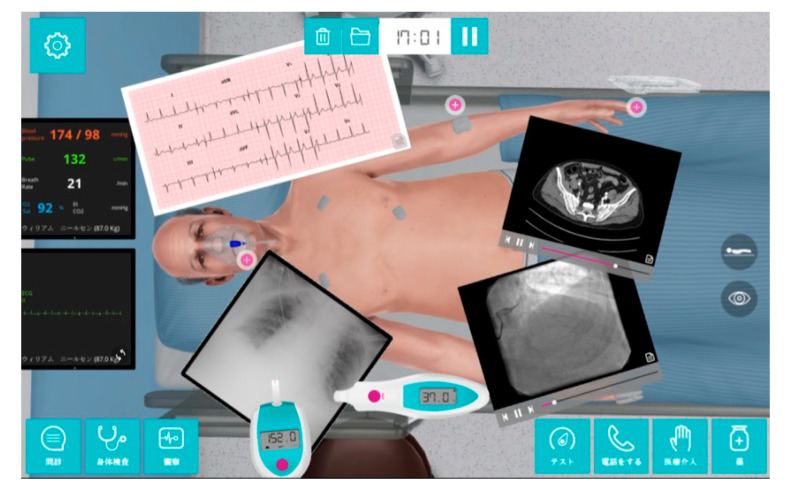
The virtual reality simulation software (^®^Body Interact, Coimbra, Portugal).

**Figure 2 ijerph-17-05325-f002:**
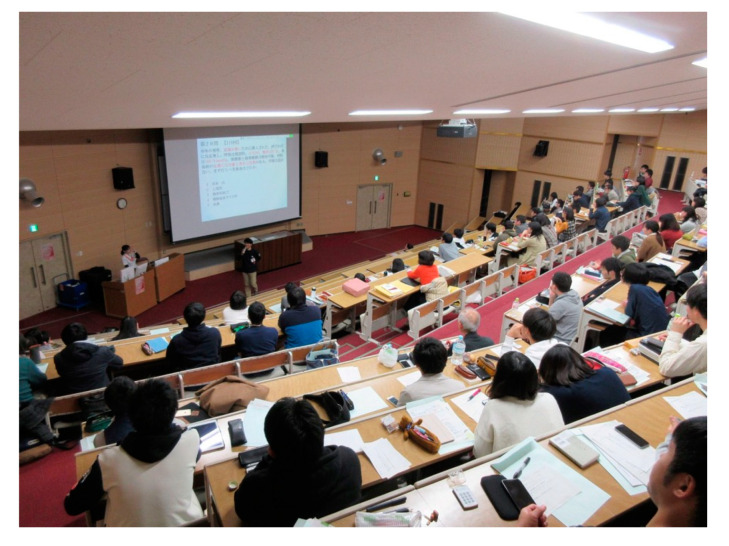
Clickers being distributed during the pre-test (March 2018).

**Figure 3 ijerph-17-05325-f003:**
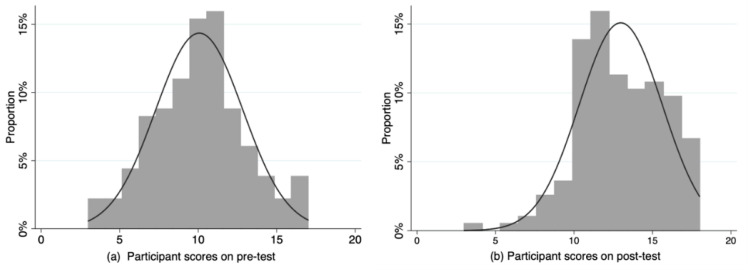
Histogram with a bell curve of participant scores (**a**) on the pre-test and (**b**) on the post-test.

**Figure 4 ijerph-17-05325-f004:**
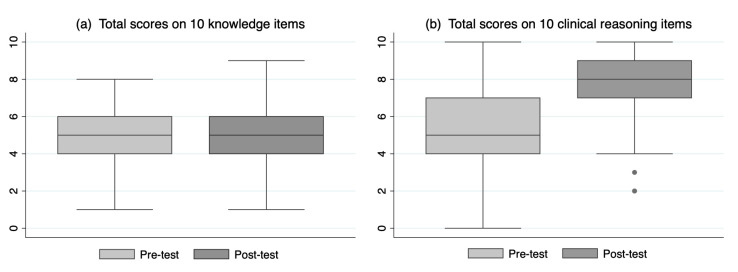
Total scores on knowledge (**a**) and clinical reasoning (**b**) items.

**Table 1 ijerph-17-05325-t001:** Pre-post test values.

Item No.	Category	Main Topic of Quiz	Pre-Test Score (*n* = 169)	Post-Test Score (*n* = 169)	Fluctuation (%)	Adjusted *p*-Value
1	CR	Management of altered mental status	25.4%	75.7%	+50.3	<0.0001 *
2	K	Electrocardiogram and syncope	52.1%	58.0%	+5.9	0.1573
3	K	Type of hormone secretion during hypoglycemia	64.5%	87.0%	+22.5	<0.0001 *
4	K	Referred pain of acute coronary syndrome	51.5%	53.8%	+2.4	0.5862
5	CR	Time course of syncope (cardiogenic)	59.2%	82.8%	+23.7	<0.0001 *
6	K	Pathophysiology of pulmonary failure	34.9%	31.4%	−3.6	0.1573
7	K	Electrocardiogram of ST elevation	34.3%	36.7%	+2.4	0.5791
8	CR	Vital signs of sepsis	73.4%	85.8%	+12.4	<0.0001 *
9	K	Anatomy of aortic dissection	55%	53.3%	−1.8	0.6015
10	CR	Management of each type of shock	66.9%	78.7%	+11.8	0.0032
11	K	Contrast CT of aortic dissection	67.5%	79.3%	+11.8	0.0016 *
12	CR	Treatment strategy of shock	56.2%	62.7%	+6.5	0.0630
13	K	Jugular venous pressure	42.6%	49.7%	+7.1	0.0455
14	CR	Differential diagnosis of hypoglycemia	24.3%	82.2%	+58.0	<0.0001 *
15	CR	Management of altered mental status	75.1%	97.0%	+21.9	<0.0001 *
16	K	Chest radiograph of heart failure	24.3%	39.6%	+15.4	<0.0001 *
17	CR	Management of syncope	53.8%	79.3%	+25.4	<0.0001 *
18	K	Symptoms of hypoglycemia	51.5%	27.8%	−23.7	<0.0001 *
19	CR	Treatment of sepsis shock	47.3%	50.9%	+3.6	0.3428
20	CR	Management of chest pain	46.7%	85.8%	+39.1	<0.0001 *

Notes: K = knowledge, CR = clinical reasoning, * = statistically significant, *p*-value < 0.0025.

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
