# Peer review of "The Utility of Virtual Patient Simulations for Clinical Reasoning Education"

_ijerph, 2020, doi:10.3390/ijerph17155325_

Round 1

Reviewer 1 Report

Thank-you for presenting this work.  This paper investigates an important topic and used a novel method.  However, I think the method has too many flaws to support the inferences draw (as detailed below).  However, with further re-design, the study could be repeated in the future to present stronger data. 

Some specific feedback:

  1. The aims as stated in the Abstract, and at the end of the introduction do not match.  In the Abstract ‘learning satisfaction’ is included, but this is not measured anywhere in the paper.
  2. The link to Covid -19 in the Introduction and Discussion is inappropriate. Education benefits from a variety of technologies that support different learning styles and needs of students, regardless of the need for social distancing. 
  3. The method is flawed and inferences cannot be drawn about the success of VRSs to “improve” CR among participants. This study showed that all students improved on CR through participation in the session. But there was not even an attempt at control.  For example, if the students had been randomly divided to receive VRSs OR to discuss a written case with similar content OR traditional methods, THEN what would the difference be in scores on the MCQ quiz?  The limitation section does not acknowledge the need for control/ experimental design. 
  4. The method contains no information on the creation of the 20-item MCQs, nor their categorization into CR or K. Are the MCQs reliable and valid? Many more details need to be added so the reader can interpret the % scores in the results.
  5. The method contains very little information on the VRs technique, and the introduction does not provide enough argument to support its use. WHY would this method be advantageous?  Is VRSs  better than other face to face methods to train CR? Better than traditional methods?
  6. The statistical analyses appear to be correct.
  7. The discussion contains information that should be included in the Introduction. Refs 32, 33,
  8. The limitations section does not contain the main problems associated with the study.

Author Response

July 10, 2020

Dear Editor and reviewers:

Thank you for allowing us to revise the manuscript. The constructive suggestions and feedback provided by the reviewers have substantially improved our paper. Our revised manuscript contains modifications based on the reviewers’ suggestions.

Once again, thank you for your thorough and supportive peer review.

On behalf of the authors, yours sincerely,

Takashi Watari, MD, MCTM, MS

  1. The aims as stated in the Abstract, and at the end of the introduction do not match. In the Abstract ‘learning satisfaction’ is included, but this is not measured anywhere in the paper.

Response

Thank you for your helpful comment and we regret that our explanation in abstract was not clear. Accordingly, we have revised the following text per the advice of a native English-speaking editor:

P1, L16

This study aimed to clarify the effectiveness of VPSs for improving clinical reasoning skills among medical students, and to compare improvements in knowledge or clinical reasoning skills relevant to specific clinical scenarios.

  1. The link to Covid -19 in the Introduction and Discussion is inappropriate. Education benefits from a variety of technologies that support different learning styles and needs of students, regardless of the need for social distancing.

Response

Thank you for raising these important points. We completely agree with the reviewers’ comments and have revised the text following these suggestions. We have removed all text that refers to COVID-19 and significantly revised the entire manuscript. All changes are marked in yellow.

  1. The method is flawed and inferences cannot be drawn about the success of VRSs to “improve” CR among participants. This study showed that all students improved on CR through participation in the session. But there was not even an attempt at control. For example, if the students had been randomly divided to receive VRSs OR to discuss a written case with similar content OR traditional methods, THEN what would the difference be in scores on the MCQ quiz? The limitation section does not acknowledge the need for control/ experimental design.

Response

Thank you for pointing out this important issue. We agree with you and we have simplified the logic. This is not a comparison between VR and other educational methods, but purely a simulated scenario. Engaging in the VPS program will be a test of whether clinical reasoning skills grow in comparison to knowledge skills. To this end, we have made significant changes, as suggested by the reviewers and we added this point as a limitation to the discussion. The changes are marked in yellow.

1) This study used simulation to compare gaining knowledge with developing clinical reasoning skills. Hence, we mentioned that it is not superior to traditional teaching methods.

2) In the limitation section, we mentioned the need for future research to elucidate the effectiveness of VPS.

Changes (Limitations)

P8 L227

“Although we attempted to ensure a uniform study setting, there are some limitations to our study. First, this is not an experiment comparing VPS to other teaching methods. For that reason, we cannot say whether it is better than traditional teaching methods, such as lectures and case discussions. This discussion point is important and further comparative experiments need to be conducted.”

  1. The method contains no information on the creation of the 20-item MCQs, nor their categorization into CR or K. Are the MCQs reliable and valid? Many more details need to be added so the reader can interpret the % scores in the results.

Response

Thank you for your helpful suggestion. We apologize for our lack of description, which makes the text difficult to understand. The 20 questions have been added as an appendix. All of these questions were taken directly from the Japanese National Medical Practitioners Qualifying Examination, so they have a certain amount of validity. Additionally, the 1st and 3rd authors have chosen to carefully divide the questions into knowledge and clinical reasoning categories. However, in the discussion section, we briefly explain that the validity of the quizzes is not perfect. 

Changes

P2 L90 “ we prepared a 20-item multiple-choice question (MCQ) quiz, which included 10 knowledge items and 10 clinical reasoning items for each of two scenarios (see Appendix 2). We categorized the questions into knowledge items and clinical reasoning items from the past questions of the Japanese National Medical Practitioners Qualifying Examination in Japan, which were relevant to each scenario. The items were randomly arranged and selected from the Japanese National Medical Practitioners Qualifying Examination. The same MCQ quiz was administered to participants pre-study and post-study to evaluate the VPS outcomes. We explained that the participants could not discuss the answers with others.

  1. The method contains very little information on the VRs technique, and the introduction does not provide enough argument to support its use. WHY would this method be advantageous? Is VRSs better than other face to face methods to train CR? Better than traditional methods?

Response

Thank you for pointing out these important issues. We agree with you and have simplified the logic. This is not a comparison between VR and other educational methods, but purely a simulated scenario. Engaging in the VPS program will be a test of whether clinical reasoning skills grow in comparison to knowledge skills. To this end, we have made significant changes, as suggested by the reviewers. The changes are marked in yellow.

Changes

P2 L46 (Introduction)

“The improved skills include procedural skills, a mixture of procedural and team skills, and some clinical reasoning skills. Moreover, virtual patient simulations (VPSs) in abnormal clinical scenarios may provide an accelerated breakthrough in improving clinical reasoning education. However, there is inadequate evidence of the usefulness of VPSs for improving clinical reasoning skills for undergraduate medical students without lecturer supervision. To the best of our knowledge, few studies have used VPSs to comparatively measure which areas of relevant case knowledge and clinical reasoning skills in symptomatology lead to better outcomes. Thus, this study had two objectives: 1) to clarify the effectiveness of VPSs for developing clinical reasoning skills among medical students without lecturer supervision, and 2) to elucidate whether VPSs improve clinical reasoning skills or knowledge of a particular case. To address these objectives, we used a pre–post experimental design to study a VPS.”

P2 L77

“This was a single center pre–post study of the effects of VPSs on clinical reasoning. A VPS software program (®Body Interact, Portugal) was used as an experimental intervention in both two-hour classes using an experimental design (see appendix 1; two scenarios were prepared). As the VPS in this study was based on a screen rather than a headset, it can be considered a partial VPS. In this software (Figure 1), the virtual patient shows the dynamic pathophysiological response to user decisions. Participants can act as real clinicians, ordering almost any tests or treatments as needed.”

  1. The discussion contains information that should be included in the Introduction. Refs 32, 33. And The limitations section does not contain the main problems associated with the study.

Response

Thank you for your constructive suggestions. The introduction has been substantially revised in response to your suggestion.

See, P1, L33 Introduction and P7, L227 Limitations.

----------------------------------------------------------------------------------------------------------

Once again, thank you for your constructive and fruitful feedback on our paper. I hope you find the revised manuscript suitable for publication.

Reviewer 2 Report

The introduction is formulated in too harsh a manner ("no interactivity," "lectures are tedious" and so on). There are numerous medical schools which succeed in conducting their lectures and clinical exercises in a dynamic and interesting manner. While it is understandable that the authors intend to juxtapose the VR courses and traditional ones, the weights must be distributed properly. The statement that VR (simulation) is inevitable appears also very strong - there are other promising approaches to simulations in clinical education.

The aim of the study is clear.

The abbreviation MCQ is not defined (and should be, to avoid ambiguity).

An important limitation of the study is its one-shot nature, whereas in clinical education series of VR exercises with increasing complexity/difficulty are imaginable (and, probably, beneficial). This should be discussed.

Otherwise, the manuscript is interesting and provides clues for shaping the future of clinical education.

Author Response

July 10, 2020

Dear Editor and reviewers:

Thank you for allowing us to revise the manuscript. The constructive suggestions and feedback provided by the reviewers have substantially improved our paper. Our revised manuscript contains modifications based on the reviewers’ suggestions.

Once again, thank you for your thorough and supportive peer review.

On behalf of the authors, yours sincerely,

Takashi Watari, MD, MCTM, MS

  1. The introduction is formulated in too harsh a manner ("no interactivity," "lectures are tedious" and so on). There are numerous medical schools which succeed in conducting their lectures and clinical exercises in a dynamic and interesting manner. While it is understandable that the authors intend to juxtapose the VR courses and traditional ones, the weights must be distributed properly. The statement that VR (simulation) is inevitable appears also very strong - there are other promising approaches to simulations in clinical education.

Response

Thanks for pointing out this issue. We have made significant revisions to the introduction and limitations section. We have also changed the text as follows.

P7 L220

“Furthermore, the greatest value of VPS is that the patient is not required to be involved in student training, and thus, there is no risk to the patient. There are also other merits, such as the ability to repeat the learning experience until the educational goal is achieved. Of course, it is not possible to complete all medical instruction through simulation education. However, we believe that if the strengths of traditional educational methods are combined appropriately with the VPS, further educational benefits can be expected.”

  1. The abbreviation MCQ is not defined (and should be, to avoid ambiguity).

Response

Thank you for your suggestion. We defined MCQ the first time it appeared in the text.

Changes

P2 L91

“…we prepared a 20-item multiple-choice question (MCQ) quiz which included 10 knowledge items and 10 clinical reasoning items for each of two scenarios (see Appendix 2). We categorized the questions into knowledge items and clinical reasoning items from the past questions of the Japanese National Medical Practitioners Qualifying Examination in Japan, which were relevant to each scenario. The items were randomly arranged and selected from the Japanese National Medical Practitioners Qualifying Examination.”

  1. An important limitation of the study is its one-shot nature, whereas in clinical education series of VR exercises with increasing complexity/difficulty are imaginable (and, probably, beneficial). This should be discussed.

Response

Thank you for your helpful suggestion. We agree and have revised the limitations significantly.

P8 L228

“Although we attempted to ensure a uniform study setting, there are some limitations to our study. First, this is not an experiment comparing VPS to other teaching methods. For that reason, we cannot say whether it is better than traditional teaching methods, such as lectures and case discussions. This discussion point is important and further comparative experiments need to be conducted. Second, the intervention should ideally be reassessed afterwards to ensure that clinical reasoning skills have indeed improved and have been retained. Further, we cannot say whether performing multiple VPS scenarios will actually develop sufficient clinical reasoning skills for a clinician. However, our study could not be followed up because the data were taken anonymously using a clicker. Third, this study was conducted in a Japanese university hospital. Since clinical medical education systems vary from country to country, there is a question of external validity. It is uncertain whether these results are applicable to other countries or institutions. Fourth, variations in the degree of participation may be possible due to the type of VR used. Further, although we tried to ensure consistency in the study setting, there may have been some unavoidable differences, such as the individual students’ active participation, concentration, and screen visibility due to seating positions. Fifth, about 10% of students were absent since no participation incentives were provided, and this may have created a selection bias. Finally, no similar studies have been conducted, and the validity of the test questions is uncertain, because they were selected from the Japanese National Medical Practitioners Qualifying Examination.”

----------------------------------------------------------------------------------------------------------

Once again, thank you for your constructive and fruitful feedback on our paper. I hope you find the revised manuscript suitable for publication.

Reviewer 3 Report

General comments Generally well written with few grammatical errors. Good background with references to back up the rationale   However, there seems to be quite a bit of mix-up of the terms "online learning" "VRS" and "traditional learning." For instance, when reading into the the methodology, I realized this is not a purely VRS study, because participants were in a traditional large lecture room with clickers. To me, it seems to be a "modified traditional lecture learning with a big screen and clickers" -  it is debatable whether this should be considered VRS. The abstract does not even briefly mention this "VRS" involves participants physically present in a large lecture hall, similar to traditional learning. And it is unclear what the authors mean by "traditional learning"- does it mean traditional lectures or traditional clinical bedside teaching?   The introduction and discussion paragraphs seem to extensively discuss online learning. However, this study is not about online learning - participants were all physically present in a large lecture room.   This study seems to underestimate the effect of influence of peers on the clicker responses. The participants were sitting very close to each other and may corroborate the same response after some informal discussion. If learning is completely online and virtual, with each participant independently giving a response, the MCQ responses could be very different.   It is not surprising that participants would at least learn something and give positive responses soon after the teaching. It can be argued that we would see similar or even superior effect with traditional learning. If the authors are trying to suggest VRS should be the new norm post COVID-19, we should see some data in this study that demonstrates virtual learning is comparable to traditional learning.   Perhaps, rather than claiming VRS can replace traditional learning, the authors could suggest "clicker with a big screen" can be a supplement to traditional learning, but this requires further studies to evaluate.   Overall, the conclusion/discussion does not seem to agree with the study data present (see more specific comments below)       Specific comments Line 96. Do you mean VRS (not SVR)   Line 135. You started using the term IQR in this line. Perhaps, define how you get the IQR in methodology/statistics section   Line 139. The Shaprio Wilk test was never mentioned in methodology/statistics section How do you interpret why post test data skewed to one side? (not mentioned in discussion)   Line 139-144.  It seems like a long, incomplete fragment, with double parenthesis on Line 142, and no closed parenthesis in Line 144. Consider re-writing it to make it clearer. If you are comparing 4 groups (2017 pre, 2017 post, 2018 pre, 2018 post) together, you cannot use McNemar. Consider ANOVA   Line 146. Figure 3 is missing   Line 159. How do you explain the decrease in the percentage of correct answers in the simple knowledge section? Could it be argued that this VRS is now making participants less knowledgeable?   Line 201. It is interesting you brought up learning retention. Did your study demonstrate whether participants retain their knowledge and clinical reasoning? If not, would you considering adding that in your limitation paragraph in Discussion?   Line 207. You noticed your research is not on a fully immersive VRS. But your abstract did not mention that. I think it would be less misleading if you can at least clarify what intervention (e.g. a screen in a large classroom, with clickers) you are using in your abstract.   Line 213-218. It can argued that traditional lecture learning also does not involve real patients or cause any harm to patients. So why is VRS superior to traditional lectures? Are you trying to list the advantages of VRS versus clinical bedside teaching? If yes, please show us the data.   Line 219-222. I think it is debatable whether the VRS in this study demonstrated cost saving effect. There is actually additional cost of hiring technical support, even though you save cost in minimizing the number of lecturers involved. As seen in this study, there is actually additional cost of having this VRS program in a traditional lecture. Please list the cost involved in this study if the authors want to claim this VRS is cost saving. And without this VRS, the lecturer would still be teaching on their own (rather than hiring additional staff). You also mentioned in Line 193 "traditionally styled lessons
to many medical students by a single experienced teacher." So why is this VRS cost saving? It is debatable whether faculty members could then "concentrate more on research and clinical practice". If the aim of VRS is saving cost, one faculty member would then be stretched to facilitate multiple VRSs, giving them less time in research and clinical practice.   Perhaps include your list of MCQs in the supplement

Author Response

July 10, 2020

Dear Editor and reviewers:

Thank you for allowing us to revise the manuscript. The constructive suggestions and feedback provided by the reviewers have substantially improved our paper. Our revised manuscript contains modifications based on the reviewers’ suggestions.

Once again, thank you for your thorough and supportive peer review.

On behalf of the authors, yours sincerely,

Takashi Watari, MD, MCTM, MS

General comments

Generally well written with few grammatical errors. Good background with references to back up the rationale   However, there seems to be quite a bit of mix-up of the terms "online learning" "VRS" and "traditional learning." For instance, when reading into the the methodology, I realized this is not a purely VRS study, because participants were in a traditional large lecture room with clickers. To me, it seems to be a "modified traditional lecture learning with a big screen and clickers" -  it is debatable whether this should be considered VRS.

Response

Thank you for pointing this out, and we apologize for the misunderstanding. Our goal is to simply present the simulation scenario without teaching, and confirm whether this form of education improves students’ clinical reasoning skills compared to knowledge skills. Because we are not comparing it to traditional learning methods, we cannot actually explain it logically. Further, as you have pointed out, this is not a full immersion VRS with a head set. We explained this point in the discussion (P7, L210). The manuscript has been substantially changed, particularly the introduction and limitations. Changes are shown in yellow.

The abstract does not even briefly mention this "VRS" involves participants physically present in a large lecture hall, similar to traditional learning. And it is unclear what the authors mean by "traditional learning"- does it mean traditional lectures or traditional clinical bedside teaching?  

Response:

Thank you for your helpful comments about the abstract and we apologize for the confusion. In accordance with the Reviewer's opinion we have significantly revised the abstract and the entire manuscript. All changes are marked in yellow.

Changes to the abstract

“Virtual Patient Simulations (VPSs) have been cited as a novel learning strategy, but there is little evidence that VPSs yield improvements in clinical reasoning skills and medical knowledge. This study aimed to clarify the effectiveness of VPSs for improving clinical reasoning skills among medical students, and to compare improvements in knowledge or clinical reasoning skills relevant to specific clinical scenarios. We enrolled 210 fourth-year medical students in March 2017 and March 2018 to participate in a real-time pre-post experimental design conducted in a large lecture hall by using a clicker. A VPS program (®Body Interact, Portugal) was implemented for one two-hour class session using the same methodology during both years. A pre–post 20-item multiple-choice questionnaire (10 knowledge and 10 clinical reasoning items) was used to evaluate learning outcomes. A total of 169 students completed the program. Participants showed significant increases in average total post-test scores, both on knowledge items (pre-test: median = 5, mean = 4.78, 95% CI [4.55–5.01]; post-test: median = 5, mean = 5.12, 95% CI [4.90–5.43]; p-value = 0.003) and clinical reasoning items (pre-test: median = 5, mean = 5.3 95%, CI [4.98–5.58]; post-test: median = 8, mean = 7.81, 95% CI [7.57–8.05]; p-value <0.001). Thus, VPS programs could help medical students improve their clinical decision-making skills without lecturer supervision.” 

P 1 L33 (Introduction)

“Traditional lectures in Japanese medical schools have been primarily conducted as didactic lectures in large classrooms, with little interactivity.”

The introduction and discussion paragraphs seem to extensively discuss online learning. However, this study is not about online learning - participants were all physically present in a large lecture room.   This study seems to underestimate the effect of influence of peers on the clicker responses. The participants were sitting very close to each other and may corroborate the same response after some informal discussion. If learning is completely online and virtual, with each participant independently giving a response, the MCQ responses could be very different.  

Response

We would like to thank the reviewer for these kind and helpful comments. We apologize for not providing sufficient description in the introduction and discussion. However, thanks to the reviewer’s constructive suggestions, we have made the necessary changes to the manuscript.

First, we revised the introduction and discussion portions about online learning. Additionally, we mentioned that the participants were in the same room.

However, before and after the MCQ, we explained that the participants could not discuss the answers with others. And, in fact, the students did not discuss with one another. The reasons were that (1) they took the quiz seriously because the questions were taken from the Japanese National Medical Practitioners Qualifying Examination, and (2) the time limit for each quiz was short and did not allow for discussion time with others.

It is not surprising that participants would at least learn something and give positive responses soon after the teaching. It can be argued that we would see similar or even superior effect with traditional learning. If the authors are trying to suggest VRS should be the new norm post COVID-19, we should see some data in this study that demonstrates virtual learning is comparable to traditional learning.  

Response

Thank you for your comment and we apologize for the misunderstanding. However, the intention we wanted to convey in this study was to see if engaging with the VPS would help develop clinical reasoning skills compared to knowledge. If we let the students solve a problem and then tell them the answer, their knowledge grows as a matter of course. Likewise, their clinical reasoning skills also increase. We assessed whether they could develop their clinical reasoning skills without the influence of direct instruction.

Due to the feasibility of the study, we were only able to adopt a pre-post design. Ideally, an RCT or two arm comparison study would be suitable for making a comparison with traditional teaching methods. Thus, we added this new limitation to the discussion. We have also added a table of student feedback to the Appendix.

Perhaps, rather than claiming VRS can replace traditional learning, the authors could suggest "clicker with a big screen" can be a supplement to traditional learning, but this requires further studies to evaluate.  

Response

Thank you for your helpful comments. As you pointed out, we used virtual reality simulation in this study. We measured whether engaging with the clinical cases on the screen leads to improved clinical reasoning skills in practice compared to knowledge. As we have attached to the appendix, this VRS is completely headset-based and not an immersion-type VRS. The product itself is a screen-based virtual reality program. For that reason, it may not be VRS in the narrow sense of the word. We corrected the discussion section in response to your point.

Changes

P2 L80

“As the VPS in this study was based on a screen rather than a headset, it can be considered a partial VPS. In this software (Figure 1), the virtual patient shows the dynamic pathophysiological response to user decisions. Participants can act as real clinicians, ordering almost any tests or treatments as needed.”

P7 L211

“In this study, since we implemented a non-immersive simulation without a virtual reality (VR) head set, we cannot consider it a true VR simulation by narrow definition. However, we gathered the participants in a large lecture hall with a shared screen to ensure uniformity of participant experience and to minimize confounding factors as much as possible. A fully immersive VR simulation, using a VR headset, may have a higher educational effect than a large shared screen. It is not easy, however, for all participants to simultaneously use a fully immersive VR simulation, and it currently requires a tremendous amount of money and high-tech equipment.”

Line 96. Do you mean VRS (not SVR) 

Response

In response to feedback from other reviewers, we have changed the terminology from VRS (virtual reality simulation) to VPS (virtual patient simulation) throughout the manuscript.

Line 135. You started using the term IQR in this line. Perhaps, define how you get the IQR in methodology/statistics section  

Response

Thank you for your suggestion. We added this explanation in response to the reviewer's point.

Changes

P4, L119

“We used an interquartile range (IQR) along with the 25th and 75th percentiles to indicate skewed data.”

Line 139. The Shaprio Wilk test was never mentioned in methodology/statistics section How do you interpret why post test data skewed to one side? (not mentioned in discussion)  

Response

We thank the reviewer for this pertinent comment. Accordingly, we have changed the following text.

Changes

P4, L120

“The Shapiro-Wilk test was implemented to examine the total score of pre–post test distributions.”

And

P6, L172

“As expected, overall scores were higher after the intervention. This can be explained by the fact that the histogram of overall scores was significantly skewed towards the higher scores. However, as a general rule with certain educational interventions, total scores will always rise immediately afterwards. This study assessed CR outcomes by using a VPS to train participants on diagnosing two clinical scenarios.”

Line 139-144.  It seems like a long, incomplete fragment, with double parenthesis on Line 142, and no closed parenthesis in Line 144. Consider re-writing it to make it clearer.

Response

Thank you for pointing out these errors, and we apologize for any confusion. We have corrected the text as follows.

Changes (Results)

P4 L134

“Compared with pre-test baseline scores, participating students showed significant increases in average total post-test scores (p-value <0.001). The total pre-test scores showed a median of 10 IQR (8–12), a mean of 10.1, and 95% CI (9.6–10.5); the post-test scores showed a median of 13 IQR (11–15), a mean of 13.0, and 95% CI (12.6–13.4). Figure 3 presents histograms of the pre-test and post-test scores for the 169 participants. The pre-test scores showed a normal distribution, while the post-test scores after the VPS intervention skewed to the right (Shapiro–Wilk test, p-value <0.022). Furthermore, no significant differences were observed for these fourth-year medical students between the years 2017 (pre-test scores: median = 10 IQR [8–11], mean = 9.7, 95% CI [9.1–10.3]; post-test scores: median = 13, IQR [11–15], mean = 12.7, 95% CI [12.1–13.2]) and 2018 (pre-test scores: median = 10 IQR [9–12], mean = 10.4, 95% CI [9.9–10.3]; post-test scores: median = 13 IQR [12–16], mean = 13.3, 95% CI [12.7–13.9]).”

If you are comparing 4 groups (2017 pre, 2017 post, 2018 pre, 2018 post) together, you cannot use McNemar. Consider ANOVA  

Response

Thank you for your comment. We apologize that our explanation was not clear. Table 1 shows the contents of McNemar's test with Bonferroni correction. As you know, the McNemar’s test should be used for a pre-post test with paired nominal data and with a dichotomous trait. This test cannot be used for continuous variables. In addition, we have not compared the four groups (2017 pre, 2017 post, 2018 pre, and 2018 post), but rather the total pre and total post scores as being the same condition. Our school has only about 100 students in each year, so we had no choice but to assess them over two consecutive years in order to increase our sample. We believe that the Wilcoxon signed-rank test is suitable for the paired pre-post test with skewed continuous variable.

Line 146. Figure 3 is missing  

Response

Thank you for pointing out the missing figure. You are correct, the figure had been moved elsewhere. We have fixed this problem.

Line 159. How do you explain the decrease in the percentage of correct answers in the simple knowledge section? Could it be argued that this VRS is now making participants less knowledgeable?  

Response

Thank you for your thoughtful comment. We apologize for our lack of explanation. We have added the following text to the discussion.

Changes (Discussion)

P7 L188

“Furthermore, the knowledge acquired in this pre-clinical stage may still be perceived as complex when based only on engagement with the VPS. For example, the scores for some of the K items decreased in the opposite direction (items 9 and 18). One possible reason may be that analyzing the chest pain and altered mental status cases through the VPS program caused learner bias due to confusion over each scenario, and the answers were pulled in an unexpected direction.”

Line 201. It is interesting you brought up learning retention. Did your study demonstrate whether participants retain their knowledge and clinical reasoning?

If not, would you considering adding that in your limitation paragraph in Discussion?  

Response

We believe that retention of knowledge and clinical reasoning skills should be further evaluated. However, our research methods are anonymous and can only be tied together using a clicker within this class only. Hence, this evaluation could not be conducted. The following has been added in response to your suggestion.

Changes (Limitations)

P8 L232

“Second, the intervention should ideally be reassessed afterwards to ensure that clinical reasoning skills have indeed improved and have been retained. Further, we cannot say whether performing multiple VPS scenarios will actually develop sufficient clinical reasoning skills for a clinician. However, our study could not be followed up because the data were taken anonymously using a clicker.”

Line 207. You noticed your research is not on a fully immersive VRS. But your abstract did not mention that. I think it would be less misleading if you can at least clarify what intervention (e.g. a screen in a large classroom, with clickers) you are using in your abstract.  

Response

Thank you for your helpful suggestion. We corrected the abstract as follows.

Changes (Abstract)

P1 L15

“Virtual Patient Simulations (VPSs) have been cited as a novel learning strategy, but there is little evidence that VPSs yield improvements in clinical reasoning skills and medical knowledge. This study aimed to clarify the effectiveness of VPSs for improving clinical reasoning skills among medical students, and to compare improvements in knowledge or clinical reasoning skills relevant to specific clinical scenarios. We enrolled 210 fourth-year medical students in March 2017 and March 2018 to participate in a real-time pre-post experimental design conducted in a large lecture hall by using a clicker. A VPS program (®Body Interact, Portugal) was implemented for one two-hour class session using the same methodology during both years. A pre–post 20-item multiple-choice questionnaire (10 knowledge and 10 clinical reasoning items) was used to evaluate learning outcomes. A total of 169 students completed the program. Participants showed significant increases in average total post-test scores, both on knowledge items (pre-test: median = 5, mean = 4.78, 95% CI [4.55–5.01]; post-test: median = 5, mean = 5.12, 95% CI [4.90–5.43]; p-value = 0.003) and clinical reasoning items (pre-test: median = 5, mean = 5.3 95%, CI [4.98–5.58]; post-test: median = 8, mean = 7.81, 95% CI [7.57–8.05]; p-value <0.001). Thus, VPS programs could help medical students improve their clinical decision-making skills without lecturer supervision.”

Line 213-218. It can argued that traditional lecture learning also does not involve real patients or cause any harm to patients. So why is VRS superior to traditional lectures? Are you trying to list the advantages of VRS versus clinical bedside teaching? If yes, please show us the data.  

Response

Thank you for your helpful comments. We acknowledge that we did not explain the issue sufficiently, and we certainly did not intend to argue this point. Indeed, logically, we believe that our results do not allow us to mention the following two points: 1) teaching with VPS is better than traditional education, and 2) traditional education is not safer. We agree with your comments, and in response to this feedback, we have made significant changes to the introduction and discussion sections. We changed the discussion points and the limitations in particular, as shown in yellow.

Changes

P7 L220

“Furthermore, the greatest value of VPS is that the patient is not required to be involved in student training, and thus, there is no risk to the patient [2,13,33,37,38]. There are also other merits, such as the ability to repeat the learning experience until the educational goal is achieved. Of course, it is not possible to complete all medical instruction through simulation education. However, we believe that if the strengths of traditional educational methods are combined appropriately with the VPS, further educational benefits can be expected.”

P8 L228

“Although we attempted to ensure a uniform study setting, there are some limitations to our study. First, this is not an experiment comparing VPS to other teaching methods. For that reason, we cannot say whether it is better than traditional teaching methods, such as lectures and case discussions. This discussion point is important and further comparative experiments need to be conducted.”

Line 219-222. I think it is debatable whether the VRS in this study demonstrated cost saving effect. There is actually additional cost of hiring technical support, even though you save cost in minimizing the number of lecturers involved. As seen in this study, there is actually additional cost of having this VRS program in a traditional lecture. Please list the cost involved in this study if the authors want to claim this VRS is cost saving. And without this VRS, the lecturer would still be teaching on their own (rather than hiring additional staff). You also mentioned in Line 193 "traditionally styled lessons to many medical students by a single experienced teacher." So why is this VRS cost saving? It is debatable whether faculty members could then "concentrate more on research and clinical practice". If the aim of VRS is saving cost, one faculty member would then be stretched to facilitate multiple VRSs, giving them less time in research and clinical practice. 

Response

Thank you for pointing out this issue. We agree with your suggestion and have removed the text you pointed out (see L211-245). Also, the discussion and introduction have been significantly revised. The changes are marked in yellow.

Perhaps include your list of MCQs in the supplement

Response

Thank you for your suggestion. I have attached the MCQ list as you pointed out.

----------------------------------------------------------------------------------------------------------

Once again, thank you for your constructive and fruitful feedback on our paper. I hope you find the revised manuscript suitable for publication.

Reviewer 4 Report

The manuscript is well written and organized reasonably. It attempts to address enhancement of clinical reasoning skills following VR-based education. There are several errors in this manuscript in verbiage and reasoning I would like to see addressed.

-It is unclear to me if the knowledge tests were shown to actually be significantly different. It is states at the start that they are, then later said to be "increased statistically", which is not necessarily the same thing, then the data is displayed in a box graph where the pre-and post data look nearly identical. I find it hard to believe, given the presentation, that the K values significantly differed.

-At several points the study is described as "longitudinal" but this is simply inaccurate. They gather data from students before and after a 2 hour session and did the examination twice with different subjects. To be longitudinal, it would imply that the same subjects were studied over time.

-I am confused as to how the VRS played out. It was unclear in the methods whether each subject had a screen to work from or not, but was clarified later on to be a shared-screen experience. This should be spelled out more clearly in the methods section.

-On that note, as described I would not evaluate this activity as being virtual reality in any way. This was at best a group activity that was simulation-based, without any of the hallmark personal instantiation that defines VR. Electronic simulations are valid activities to explore, but the validity of comparing this to a VR simulation is extremely off-base. Even if we were to accept the "partial VRS" the authors state, that does not necessarily apply to a full VRS experience. There is value in examining electronic simulations as classroom activities, but to call this VR is disingenuous.

To be accepted, I would suggest the paper be re-written to be more specific and accurate in what it is describing and what your data actually means.

Author Response

July 10, 2020

Dear Editor and reviewers:

Thank you for allowing us to revise the manuscript. The constructive suggestions and feedback provided by the reviewers have substantially improved our paper. Our revised manuscript contains modifications based on the reviewers’ suggestions.

Once again, thank you for your thorough and supportive peer review.

On behalf of the authors, yours sincerely,

Takashi Watari, MD, MCTM, MS

-It is unclear to me if the knowledge tests were shown to actually be significantly different. It is states at the start that they are, then later said to be "increased statistically", which is not necessarily the same thing, then the data is displayed in a box graph where the pre-and post data look nearly identical. I find it hard to believe, given the presentation, that the K values significantly differed.

Response

Thank you for pointing out this important issue. We felt the same way when considering the data visually, but this is the actual result. We apologize for any confusion this may have caused to the reader. Perhaps the reviewer was confused by the way the numerical data was presented in terms of mean and median. The box plot has the same median value, so visually, it looks almost unchanged. We have therefore added box plots for both (Figure 4, L168).

A pre-post comparison of the total score indicates that it statistically increased. However, whether this is statistically significant in terms of medical education is another matter. On the other hand, just engaging with the VPS is a dramatic increase as far as CR is concerned, but the K score did not improve much even though the students solved the same items.

-At several points the study is described as "longitudinal" but this is simply inaccurate. They gather data from students before and after a 2 hour session and did the examination twice with different subjects. To be longitudinal, it would imply that the same subjects were studied over time.

Response

Thank you for your helpful comments, and we believe that you are absolutely correct. We have changed the study approach from "longitudinal" to pre-post. Thank you for these important suggestions which have allowed us to make improvements to our paper.

-I am confused as to how the VRS played out. It was unclear in the methods whether each subject had a screen to work from or not, but was clarified later on to be a shared-screen experience. This should be spelled out more clearly in the methods section.

Response

Thank you for these helpful comments. We have added to the methods section that this product is screen-based.

Changes

P2 L81

“As the VPS in this study was based on a screen rather than a headset, it can be considered a partial VPS. In this software (Figure 1), the virtual patient shows the dynamic pathophysiological response to user decisions. Participants can act as real clinicians, ordering almost any tests or treatments as needed.”

-On that note, as described I would not evaluate this activity as being virtual reality in any way. This was at best a group activity that was simulation-based, without any of the hallmark personal instantiation that defines VR. Electronic simulations are valid activities to explore, but the validity of comparing this to a VR simulation is extremely off-base. Even if we were to accept the "partial VRS" the authors state, that does not necessarily apply to a full VRS experience. There is value in examining electronic simulations as classroom activities, but to call this VR is disingenuous. To be accepted, I would suggest the paper be re-written to be more specific and accurate in what it is describing and what your data actually means.

Response

Thank you for your thoughtful comments and suggestions on this point. The term is still undefined at this stage. It's a challenging question, but I understand very well what the reviewers are saying. This product is a screen-type product, so it is not the same as the headset-type. If you decide that it is not appropriate to use the term "virtual reality," in that sense, we must change our definition of the word. We've been discussing this issue, and consider the program to be a virtual reality simulation. However, I believe we should use "virtual patient simulation" rather than "virtual reality simulation." Based on our consideration of this issue, we revised our manuscript significantly.

----------------------------------------------------------------------------------------------------------

Once again, thank you for your constructive and fruitful feedback on our paper. I hope you find the revised manuscript suitable for publication.

Round 2

Reviewer 1 Report

Many thanks for sending the revised manuscript.

I find the paper much improved and appreciate the authors' attention to  detail in the revisions provided.  

I am very satisfied by all the changes and believe they address all my concerns.

I was looking forward to viewing the Appendix with the 20-item quiz, but note it was not attached.,  i hope it can be published for the on-line version (that it does not breach any copyright?)

Reviewer 3 Report

Thanks for considering the suggestions from the reviewers. The only other suggestions I want to make is that on Line 29, rather than saying "without lecturer supervision," consider "with minimal supervision." It would be a more accurate description because your lecture did have a lecturer present. It would almost be impossible to run a lecture without any facilitator/lecturer.

Reviewer 4 Report

I appreciate the authors taking the time to consider my suggestions. The box plots are far more understandable now as being significant.

My major point about the simulation not being virtual reality was addressed adequately by changing the context to virtual patient, which is accurate for the intervention. It seems like there is still some confusion in the definition, at least given the author's comments, but I wonder if some of that is somewhat due to translational differences. VR as it is defined in the serious game space requires an element of bodily immersion, most often obtained through headset displays, but can be done in other ways.  I think that you made a good call in reworking it to be Virtual Patient, and I commend the work.